# Prevalence of Asymptomatic Malaria Infections in Seemingly Healthy Children, the Rural Dzanga Sangha Region, Central African Republic

**DOI:** 10.3390/ijerph18020814

**Published:** 2021-01-19

**Authors:** Krzysztof Korzeniewski, Emilia Bylicka-Szczepanowska, Anna Lass

**Affiliations:** 1Department of Epidemiology and Tropical Medicine, Military Institute of Medicine, 128 Szaserów St., 04-141 Warsaw, Poland; 2Department of Occupational, Metabolic and Internal Diseases, Institute of Maritime and Tropical Medicine, Medical University of Gdańsk, 9B Powstania Styczniowego St., 81-519 Gdynia, Poland; 34th Department of Infectious Diseases, Provincial Hospital for Infectious Diseases, 37 Wolska St., 01-201 Warsaw, Poland; emilia.bylicka@wp.pl; 4Department of Tropical Parasitology, Institute of Maritime and Tropical Medicine, Medical University of Gdańsk, 9B Powstania Styczniowego St., 81-519 Gdynia, Poland; anna.lass@gumed.edu.pl

**Keywords:** asymptomatic malaria, *Plasmodium falciparum*, Dzanga Sangha, Central African Republic

## Abstract

According to the World Health Organization 94% of global malaria cases and 94% of global malaria deaths have been reported from Africa. Unfortunately, it is difficult to determine the exact prevalence of disease in some African countries due to a large number of asymptomatic cases. The aim of this study was to assess the prevalence of malaria infections in seemingly healthy children living in the Central African Republic (CAR). CareStart^TM^ Malaria HRP2 rapid diagnostic test (RDT) targeting *Plasmodium falciparum* was used to test a group of 500 asymptomatic children aged 1-15 years old (330 settled Bantu and 170 semi-nomadic BaAka Pygmies) inhabiting the villages in the Dzanga Sangha region (south-west CAR) in March 2020. In total, 32.4% of asymptomatic Bantu and 40.6% of asymptomatic Pygmy children had a positive result of malaria RDT. Our findings allowed us to demonstrate the high prevalence of asymptomatic malaria infections in south-west CAR. RDTs seem to be a useful tool for the detection of *Plasmodium falciparum* in areas with limited possibilities of using other diagnostic methods, such as light microscopy and molecular biology.

## 1. Introduction

Malaria is a vector-borne parasitic disease. In humans, it is mainly caused by five protozoan species: Plasmodium falciparum, Plasmodium vivax, Plasmodium ovale (P. ovale curtisi, P. ovale wallikeri), Plasmodium malariae, and Plasmodium knowlesi (a species most often found in South-East Asia whose natural hosts are monkeys) [1,2]. Thanks to the latest developments in molecular diagnostics another two enzootic species of Plasmodium have recently been discovered, these include: P. simium and P. cynomolgi, as yet, however, very little is known of the prevalence and clinical picture of the two species [3,4]. Malaria is most often transmitted by a bite from an infective female Anopheles mosquito (infection vector) when invasive Plasmodium forms are released into the bloodstream of a human host (sporozoites are mostly deposited in the skin by the biting mosquito and need to migrate through the dermis to find and invades blood capillaries) [5]; it can also be transmitted (but relatively rarely) through transfusion of blood or vertically, from an infected mother to the fetus [1].

According to the latest World Health Organization’s (WHO) report on malaria prevalence, diagnostics, treatment, and prevention (the World Malaria Report 2020), a total of 229 million people globally fall ill with the disease each year and 94% of malaria cases are reported from Africa. The estimated number of malaria deaths is 409,000 per year, of which 94% occur in Africa [6]. The WHO predicts that due to the ongoing COVID-19 pandemic, which may potentially lead to a collapse of the national healthcare systems in many developing countries, the total number of deaths from malaria in Sub-Saharan Africa may reach 769,000 in 2020, which is similar to the mortality level last seen 20 years ago [7].

According to the WHO, *P. falciparum* is responsible for 99.7% of all malaria cases in Africa and as much as 100% of cases in sub-Saharan Africa [5]. Therefore, in line with the WHO recommendations, rapid diagnostic tests (RDTs) targeting *P. falciparum* are routinely used for the diagnosis of malaria in the region [8]. Malaria has long been a major health issue in Africa, and it accounts for 9% of all diseases reported from the continent [9]. In some African countries, e.g., in the Central African Republic (a country with a population of 4.8 million people) malaria is considered to be responsible for as much as 40% of all reported illnesses and 10% of all registered deaths, especially in the pediatric population [10]. The WHO reports of over 380,000 laboratory-confirmed malaria cases in the Central African Republic each year, all apparently caused by the *P. falciparum* species (the disease is transmitted throughout the year and in all parts of the country). In fact, the actual number of malaria cases in the CAR may be several times higher than the WHO suggests due to exceptionally high rates of asymptomatic cases [11].

The aim of the study was to assess the prevalence of asymptomatic malaria infections in sub-Saharan Africa on the example of seemingly healthy children living in the rural parts of the Dzanga Sangha region in South-West Central African Republic.

## 2. Materials and Methods

### 2.1. Study Population

The study was conducted in March 2020 (the end of dry/warm season with average temperatures ranging between 21 and 33 °C); it involved a group of 500 children of both sexes, aged 1–15 years (330 settled Bantu and 170 semi-nomadic BaAka Pygmies) living in the Monasao village and surroundings in the Dzanga Sangha region (Bayanga subprefecture, Sangha-Mbaere prefecture) in South-West Central African Republic (Figure 1).

Most Bantu people are farmers who live in villages set up along the main communication routes in the region. Their livelihood largely depends on agriculture (growing crops) but some also work in gold or diamond mining industry (the two minerals being the most valuable natural resources found in the CAR). BaAka Pygmies, on the other hand, are a semi-nomadic people who inhabit local forests, or their outskirts, mostly in the vicinity of Bantu villages. Pygmies are often economically exploited by the Bantu people and are frequently used as cheap labour force.

Monasao is a village 50 km north of the Bayanga village (the capital of the Bayanga subprefecture) and 60 km south of the Nola city (the capital of Sangha-Mbaere prefecture), which lies at the altitude of 510 m above sea level. Initially, Monasao was inhabited entirely by the Pygmy tribes. Today, however, the village with its population of around 4000 people is inhabited by both the settled Bantus and the semi-nomadic BaAka Pygmies. The latter, however, tend to build their huts on the outskirts of the village. The Pygmies survive by hunting, fishing and gathering wild plants in local forests but they are also employed for field work by some Bantu people.

Bantu and Pygmy children were recruited for the study at a health care facility run by the local catholic mission (where RDTs were the only available diagnostic tool for malaria detection) and at a local primary school in Monasao. The inclusion criteria were: age ≤15 years and absence of clinical signs and symptoms of malaria (body temperature ≤37.5 °C). The exclusion criteria were presence of malaria signs and symptoms (fever ≥37.6 °C, chills, headache, vomiting, diarrhea, joint pain, and general weakness), anti-malarial treatment received in the past 28 days and difficulties in venipuncture procedure. All participants had the right to withdraw from the study at any stage. On behalf of the pediatric patients, the consent to participate in the study was given by their parents or legal guardians, who had first been informed of the study purpose and methods. The parents/guardians provided information on the age and sex of their children. The interviews were conducted in their native language, either in Sango (a language spoken by the Bantu) or Mbenzele (the language used by the BaAka Pygmies). Medical personnel responsible for carrying out the study procedures measured the participants’ body weight (using a digital body scale) and body temperature (in the frontal or temporal part of the head) using an electronic thermometer (correct temperature was defined as ≤37.0 °C, low-grade fever: 37.1–37.5 °C, fever: ≥37.6 °C). Next, medical staff performed rapid diagnostic tests for malaria as well as hemoglobin measurements to identify any possible cases of anemia; hemoglobin level was measured using a portable analyzer.

### 2.2. Malaria Screening and Blood Sampling Procedures

Immuno-chromatographic rapid diagnostic test (RDT) targeting Pf HRP2 (*Plasmodium falciparum* histidine-rich protein 2) was used for the detection of *P. falciparum* in vitro, in a whole blood sample (CareStart^TM^ Malaria HRP2 Pf Ag, Access Bio, Somerset, NJ, USA). To perform the test a drop of capillary blood (5 µL collected from the test tube with a calibrated pipette provided by the manufacturer) is added into a sample well from which the specimen migrates through the nitrocellulose membrane (paper chromatography) by capillary action. Next, 2 drops of buffer assay attached to the test are added into the buffer well. The test pad is pre-coated with colloidal gold nanoparticles conjugated with specific antibodies. After the blood sample has been absorbed by the membrane, *P. falciparum* HRP2 antigens bind to the anti–HRP2 antibodies (if the test is positive). The reaction leads to the formation of antigen-antibody complexes which the test is able to detect. The interpretation of the test will depend on the presence or absence of colour bands in the control and test areas (the presence of the color line in the test area confirms the presence of antigens in the blood sample). Thus, the presence of two bands—in the control area and the test area indicates a positive result of the test. The presence of only one band in the control area within the result window indicates a negative result. The test is invalid and the interpretation of the test not possible if the line in the control area does not appear or it only appears in the test area. The results can be read after 20 min. The RDT sensitivity (the ability of the test to correctly identify patients who have malaria, true positive rate) is 98% and its specificity (the ability of the test to correctly identify patients who do not have malaria, true negative rate) is 97.5% (according to the manufacturer’s specifications) [12].

Asymptomatic malaria was defined as the presence of *P. falciparum* on RDT test with no clinical signs of the disease (body temperature ≤37.5 °C, no headache, chills, joint pain, weakness, vomiting or diarrhea).

### 2.3. Hemoglobin Measurements

Hemoglobin level was measured using a portable DiaSpect^TM^ analyzer (EKF Diagnostics, Penarth, UK) by examining a peripheral blood sample from a finger prick. The definition of anemia was based on the WHO criteria, where hemoglobin level of ≥11 g/dL is regarded as normal, 10.0–10.9 g/dL as mild, 7.0–9.9 g/dL as moderate, and <7.0 g/dL is regarded as severe anemia.

### 2.4. Statistical Methods

The statistical analysis was performed using StatSoft Inc. (2014) STATISTICA (data analysis software system) version 12.0. www.statsoft.com (Kraków, Poland) and Microsoft Excel. Quantitative variables were characterized by the arithmetic mean, standard deviation, median, minimum and maximum value (range), and 95% CI (confidence interval). The variables of the qualitative type were presented in terms of counts and percentages (percentage). The Shapiro-Wilk test was used to examine if quantitative variables were normally distributed in a population. Levene’s test (the Brown-Forsythe test) was used to check the assumption of equal variances. The Student’s *t*-test or Mann-Whitney U test (under conditions when the Student’s *t*-test is not applicable or for variables measured on an ordinal scale) were used to determine if the differences between the two study groups (subjects with a positive RDT result vs. subjects with a negative RDT result) were statistically significant. Chi-square tests of independence were used for qualitative variables (respectively using Yates correction for cell counts below 10, checking Cochran conditions and Fisher’s exact test). In all calculations, the level of significance was set at *p* = 0.05.

### 2.5. Ethical Approval

The research project was approved by the Ministry of Scientific Research and Technological Innovation, Bangui, Central African Republic (Research Authorization) and the Committee on Bioethics at the Military Institute of Medicine, Warsaw, Poland (Decision No. 22/WIM/2020) under the Declaration of Helsinki (1996) and the rules elaborated by the European Union “Good clinical practice for trials on medicinal products in the European Community. The rules governing medicinal products in the European Community” (1999) ratified by the Committee of Ethics in Poland (March 1993). The tests performed in the Central African Republic were carried out with the written consent of each participant and under the supervision of Father Wojciech Lula (a catholic mission superior and a manager of the healthcare center in Monasao, Bayanga subrefecture, Sangha-Mbaere prefecture), with considerable help from the medical staff working at the healthcare center and the teaching staff from the Monasao primary school.

## 3. Results

### 3.1. Characteristics of the Study Participants

In the group of Bantu children (*n* = 330), a total of 107 subjects (32.4%) tested positive on mRDT; the mean age of the children with mRDT (+) was 8.7 years and their mean body weight was 22.5 kg. The mean body temperature of the tested children, both with mRDT (+) and with mRDT (–) was 36.8 °C. Of all the Bantu children with mRDT (+), 45.8% were girls and 54.2% were boys. We have not observed any statistically significant correlations between age, body weight, body temperature, sex and the mRDT result (Table 1).

In the group of Pygmy children (*n* = 170), a total of 69 children (40.6%) tested positive on mRDTs; the mean age of the children with a positive result on mRDT was 7.0 years. Children with mRDT (+) were statistically significantly younger (*p* = 0.0139). The mean body weight of the children with mRDT (+) was 16.1 kg. Children with mRDT (+) had significantly lower body weight (*p* = 0.0038). The mean body temperature of the children with mRDT (+), as well as those with mRDT (–) was 36.8 °C. Of all the children with mRDT (+), 47.8% were girls and 52.2% were boys. No statistically significant correlation was found between the participants’ sex and their body temperature and the mRDT result (Table 2).

### 3.2. Prevalence of Asymptomatic Malaria

RDT results showed that the overall prevalence of asymptomatic malaria in the study group (*n* = 500) was 35.2%; 32.4% in Bantu children (*n* = 330) and 40.6% in Pygmy children (*n* = 170).

#### 3.2.1. Bantu Children

No statistically significant correlation was found between age, body weight, body temperature, hemoglobin level and the presence of asymptomatic malaria in 150 Bantu females.

Similarly, no statistically significant correlation was found between the age, body weight, body temperature, hemoglobin level and the presence of asymptomatic malaria in 180 Bantu males.

Regression analysis was used to determine the influence of such variables as sex, age, body weight, body temperature and hemoglobin level on the result of the mRDT in the group of Bantu children. Univariate regression analysis demonstrated that a higher hemoglobin level was associated with a lower probability of a positive result on mRDT test. Normal body temperature was associated with a lower probability of a positive result on mRDT test, whereas the presence of low-grade fever was associated with a higher probability of mRDT (+). Multivariate regression analysis demonstrated that the variables had no statistically significant effect on the probability of obtaining a positive mRDT result (Table 3).

#### 3.2.2. Pygmy Children

In the group consisting of Pygmy females (*n* = 81), the mean age of the girls with mRDT (–) was 8.4 years, and of those with mRDT (+) 6.8 years. Females with a positive result on mRDT were statistically significantly younger (*p* = 0.0013). The mean body weight of Pygmy females with mRDT (−) was 19.8 kg, whereas of those with mRDT (+) it was 16.2 kg. Females with mRDT (+) had significantly lower body weight (*p* = 0.0017). No statistically significant correlation was found between temperature, hemoglobin level and the presence of asymptomatic malaria.

No statistically significant correlations were found between age, body weight, body temperature, hemoglobin level and the presence of asymptomatic malaria in 89 Pygmy males.

Regression analysis was used to determine the influence of such variables as sex, age, body weight, body temperature and hemoglobin level on the result of the mRDT in the group of Pygmy children. The univariate regression analysis demonstrated that older age and higher body weight were associated with a lower probability of a positive result on mRDT test. The multivariate regression analysis indicated that only higher body weight was associated with a lower probability of a positive mRDT result (Table 4).

### 3.3. Prevalence of Anemia

The mean hemoglobin level in Bantu children with a positive result on mRDT was 10.3 g/dL, and in those with a negative result of mRDT it was 10.7 g/dL. Children with mRDT (+) had a significantly lower hemoglobin level (*p* = 0.0399). 13.1% of the study participants with mRDT (+) vs. 19.7% with mRDT (–) had normal hemoglobin level, which confirms a high prevalence of mild to moderate anemia in the region, regardless of the mRDT result (Table 5).

The mean hemoglobin level in Pygmy children with mRDT (+) was 9.9 g/dL, whereas in those with mRDT (–) it was 10.2 g/dL. No statistically significant correlation between hemoglobin level and the result of mRDT was found. Only 5.8% study participants with mRDT (+) vs. 13.9% with mRDT (–) had normal hemoglobin level. Similarly as in Bantu children, the prevalence of mild to moderate anemia was high among the Pygmy children, regardless of the mRDT result (Table 6).

## 4. Discussion

The study aimed to determine the occurrence of asymptomatic malaria infections in children living in the Central African Republic (CAR). According to the official statistics malaria affects approximately 8% of the country’s population each year. However, the WHO has indicated that the actual number of malaria cases in the CAR is grossly underestimated and in fact it might be even several times higher than the reports suggest. This is largely due to the fact that a significant proportion of malaria cases in the region are asymptomatic [11]. The epidemiological situation of infectious diseases (malaria, tuberculosis, and measles) continues to deteriorate in the Central African Republic, which is largely attributable to the effects of natural disasters (such as devastating floods during the rainy season) that affect the country on a regular basis. The United Nations Office for the Coordination of Humanitarian Affairs (OCHA) has estimated that in 2021 as much as 57% of the CAR residents (2.8 million) will be in need of humanitarian aid, mostly due to food shortages, lack of medical care and a high prevalence of diseases [13]. Of all the problems the country has to face, limited access to health care seems to be the most problematic and challenging. Hulland et al. [14] has pointed out that in many parts of the country (including in the south-west, where the present research project was carried out) it takes longer than 24 h to reach the nearest medical facility. In such circumstances, it is impossible to generate accurate data on actual morbidity rates in the country. The results of a recent screening that was carried out in the region revealed high prevalence of malaria among the local people. According to the data from the National Committee for Epidemic Control in Chad, 60.5% of 943,000 malaria tests performed between January and September 2020 were positive, and most positive tests were reported from the southern parts of the country, i.e., the region neighboring to the Central African Republic (Logone oriental, Mandoul, and Moyen-Chari) [15]. In countries where medical care is hard to obtain and where advanced laboratory techniques are unavailable, rapid diagnostic tests (RDTs) are often the only available tool for malaria diagnosis [16,17,18,19]. RDTs are immune-chromatographic tests which identify malaria antigens (e.g., *P. falciparum* histidine-rich-protein 2 /Pf HRP2/ or the enzyme called *Plasmodium* lactate dehydrogenase /pLDH/). RDTs have their limitations as they cannot provide quantitative results and can stay positive months after infection with *P. falciparum,* or are, if testing for pLDH, positive only while there are living parasites in the blood [20] but they have the advantage of being cheap, fast and less complicated. Their use requires very little training and expertise compared to light microscopy (if the microscopist is not well trained, there could be unacceptably high-positive or high negative results), and because RDTs do not require reagent preparation or a power supply they can easily be used during field work [21]. RDTs seem to be very useful for detecting *Plasmodium falciparum* infections, especially in areas where other diagnostic methods, such as light microscopy or molecular biology techniques are not available. The accuracy of RDTs for the diagnosis of uncomplicated *P. falciparum* infection is equal or superior to routine microscopy (but inferior to expert microscopy) [22]. Many researchers believe that the quality of microscopic diagnosis of malaria in many regions of sub-Saharan Africa is not satisfying [23,24,25]. The number of malaria RDTs used in the public sector of endemic countries has increased from <200,000 in 2005 to 88,000,000 in 2010, and an annual demand for 1.5 billion RDTs has been forecasted globally [22]. The present research project used RDTs targeting Pf HRP2 to detect *P. falciparum* in vitro, in a whole blood sample (CareStart^TM^ Malaria HRP2 Pf Ag, Access Bio, Somerset, NJ, USA). The fact that WHO recommends simplified diagnostic procedures, i.e., the use of RDTs capable of detecting only one *Plasmodium* species (*P. falciparum*) may become a major obstacle to effective malaria control in Africa. According to WHO, *P. falciparum* is the etiological factor of up to 100% malaria cases in sub-Saharan Africa, but it must be emphasized that a large proportion of cases that occur in the region are never diagnosed because of widespread asymptomatic carriage of *Plasmodium*. Unpublished findings of a study which was carried out by the authors in the Bayanga subprefecture (in the same region where the present study was conducted) in 2018, using molecular methods (PCR), in a group of 540 Pygmies (aged 1–75 years) presenting with clinical signs of malaria, confirmed the presence of *P. falciparum* in 94.8% patients, but also found genetic material of *P. malariae* in 11.1%, *P. ovale* in 9.8%, and *P. vivax* in 0.7% study subjects. The results obtained indicate that there are different species of *Plasmodium* in Central Africa, which is a significant finding with respect to implementing effective diagnostic procedures and treatment protocols (currently, it is quite common that patients with a negative result on mRDT which is specific for detecting *P. falciparum* only, do not receive antimalarial drugs although they manifest clear signs of the infection).

One should also be aware of numerous limitations of RDTs (poor sensitivity at low parasite densities, susceptibility to the prozone effect /Pf HRP2-detecting RDTs/), the possibility of false negative results (due to Pf HRP2 deficiency in the case of *pfhrp2* gene deletions /Pf HRP2-detecting RDTs/ and cross-reactions between *Plasmodium* antigens and detection antibodies), as well as the possibility of false-positive results caused by other infections [22].

The results of this study indicate that malaria is a major public health issue in Central African Republic (CAR) and malaria diagnosis remains a significant challenge for local health care professionals. The most vulnerable groups affected by malaria in high-transmission areas, such as sub-Saharan Africa, are children [26]. In malaria- endemic countries, many *Plasmodium falciparum* infections are asymptomatic [27]. The asymptomatic carriers do not seek treatment for their infection and, therefore, add up to a reservoir of parasites available for transmission by *Anopheles* mosquitoes [28]. According to some authors, long term asymptomatic carriage may represent a form of tolerance to the parasite in children building up their immune response. In this way, asymptomatic carriage would protect these children from developing a mild or severe malaria attack, by keeping their immunity effective [29,30]. Conversely, asymptomatic carriage may represent a mode of entry to symptomatic malaria, especially in young children [31].

The results of this study, which was aimed to assess the prevalence of asymptomatic malaria infections in a sample of 500 children, aged 1–15 years, living in the rural areas of the Dzanga Sangha region in the south-west of the CAR, revealed 35.2% of asymptomatic malaria cases: 32.4% in Bantu and 40.6% in Pygmy children. A total of 516 children entered the study (339 Bantu and 177 Pygmy), of which 16 were excluded because of fever ≥37.6 °C (their parents/ legal guardians did not report of any other signs and symptoms of malaria, such as chills, headache, vomiting, diarrhea, joint pain, general weakness; nor did they report of anti-malarial treatment received by their children in the past 28 days; no difficulties occurred in venipuncture procedure). Of the 16 children with fever ≥37.6 °C (9 Bantu and 7 Pygmies) who were excluded from the study, a majority presented with symptoms of a respiratory infection (cough, rhinitis); the possibility of other febrile illnesses, that are epidemic (dengue) or endemic (tuberculosis, lymphatic filariasis caused by *Wuchereria bancrofti*) in Central Africa, which have not been studied in the local population, should also be considered. The present study is the first screening ever conducted in asymptomatic patients with malaria infection in the Dzanga Sangha region, in South- West CAR.

Our findings in this respect are similar to those reported by Ndamukong-Nyanga et al. in the neighboring Cameroon. The study by Ndamukong-Nyanga et al., which involved a group of over 400 children and used the same CareStart^TM^ Malaria HRP2 Pf Ag RDTs, which were used in our study, revealed that 27.9% of the participants had malaria [32]. Screening tests for malaria with RDTs by Maziarz et al. [33] in northern Uganda in a sample of more than 1,000 seemingly healthy children under 16 years old identified asymptomatic malaria infections in 52.4% participants. Asymptomatic infections go unnoticed and thus are never treated [34], resulting in anemia, disturbed concentration and learning difficulties, which eventually lead to school absenteeism [35]. They are prevalent in all parts of sub-Saharan Africa [36,37,38] and pose a real challenge for malaria prevention and control strategies on the continent. Most malaria data are based on national surveys, which are not sufficiently granular to provide local insights [39], they tend to focus on participants in a restricted age range, usually children [40], or on clinical cases [41], which provide limited information about the asymptomatic cases that represent the main reservoir of infection that spawns new clinical cases.

The mean age of asymptomatic children with a positive result on mRDT in our research study was 8.7 years in the Bantu children and 7.0 years in Pygmy children. Pygmy children with mRDT (+) were significantly younger, we also found that older age and higher body weight were associated with a lower probability of a positive result on mRDT in this ethnic group. Similar findings have been reported by Ndamukong-Nyanga et al. [32], where the highest occurrence of malaria was reported among children aged between 6 and 10 years old. According to these authors, children aged >10 years are generally more knowledgeable about malaria and are more likely to take preventive measures against the infection compared to the younger children. The drop in occurrence of malaria with an increase in age could also be related to the acquisition of protective immunity due to repeated infections as children grow older in high transmission areas [42]. These findings have been confirmed by the results of a study conducted in Tanzania which showed a reduction in positive mRDTs among older children [43].

We have not found any statistically significant correlation between sex and the result of the mRDT in our study sample. In the group of Bantu children, boys accounted for 54.2% of children with mRDT (+) and for 54.7% of those with a negative result on mRDT; in the group of Pygmy children, boys accounted for 52.2% of those with a positive result on mRDT and 52.5% of the children with mRDT (–). A study by Teh et al. [27] and Kimbi et al. [42] carried out in Cameroon show different results, they found that malaria occurrence was higher in girls than in boys, probably due to the fact that females spend more time outdoors at dusk and dawn performing household chores, and are therefore more exposed to mosquito bites. The mean body temperature in asymptomatic Bantu and Pygmy children with mRDT (+) was found to be normal (36.8 °C), the same as in children with mRDT (–). We found that the presence of low-grade fever (≤37.5 °C) increased the probability of obtaining a positive result on mRDT test. The mean hemoglobin level in asymptomatic Bantu children was found to be 10.6 g/dL (range 5.1–14.7) vs. 10.1 (range 5.4–13.8) in asymptomatic Pygmy children. Our study has confirmed a high occurrence of mild to moderate anemia in both ethnic groups. Bantu children with a lower hemoglobin level were found to be more likely to have mRDT (+). These findings are comparable with the results obtained by Teh et al. [27], who found that there were more positive mRDTs among anemic than non-anemic children. Common occurrence of anemia has also been confirmed by a cross-sectional study by Nzombo et al. The study, which aimed to assess the occurrence of malaria in a sample of more than 300 asymptomatic Tanzanian school-children aged 6–13 years, revealed that their mean hemoglobin level was 10.1 g/dL [44].

The strength of the presented study was the assessment of malaria prevalence in asymptomatic children aged 1–15 years old living in remote and hard-to-reach parts of the Central African Republic. Because asymptomatic individuals are not usually tested for malaria, it is impossible to assess the actual malaria morbidity in the country. The study had its limitations as well. Participants were recruited from among the children population living in the Monasao village or in its close vicinity, which limited the sample size. Further, limiting the sample to pediatric population only (the total population of Bayanga subprefecture has been estimated at 12,000 people) prevented the assessment of malaria prevalence in the adult population, especially in pregnant women and patients with underlying conditions.

## 5. Conclusions

The overall occurrence of asymptomatic malaria in children living in the south-western parts of the Central African Republic is very high. RDTs seem to be an effective method for detecting *Plasmodium falciparum* infections, especially in areas where the access to medical care is limited and where other diagnostic methods, such as light microscopy or molecular biology are unavailable. The WHO has warned that there may be an increase in malaria-related morbidity and mortality in Africa in the next few years, because the resources which are at the disposal of national health care systems will, in the first place, be allocated to finance the COVID-19 control strategies, rather than managing the spread of tropical diseases. Given the fact that the Bayanga sub-prefecture in the south-west of the Central African Republic is one of the most attractive tourist destinations in Africa (it is where the Dzanga Sangha National Park with its famous lowland gorillas is located), the knowledge of the exact occurrence rates of malaria in sub-Saharan Africa seems to be as important for the local health care providers as for the centers for travel and tropical medicine operating in developed countries on other continents.

## Figures and Tables

**Figure 1 ijerph-18-00814-f001:**
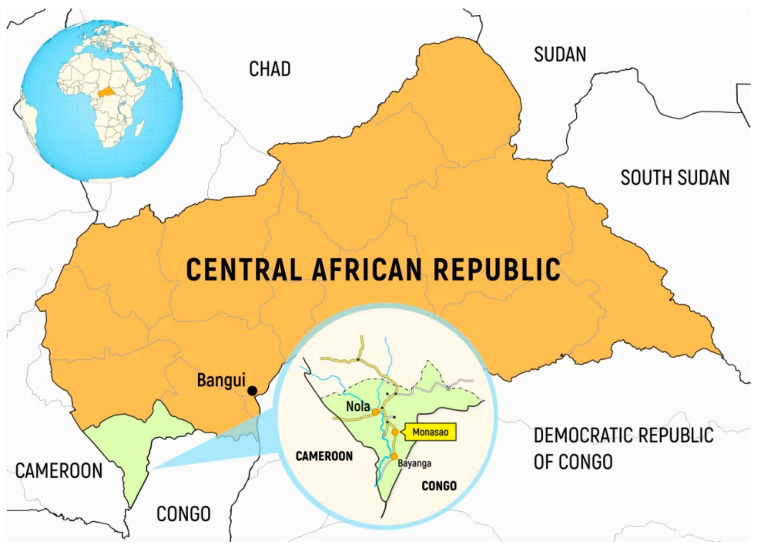
Map of South-West Central African Republic, Sangha-Mbaere prefecture.

**Table 1 ijerph-18-00814-t001:** Characteristics of Bantu children (*n* = 330).

Variables	mRDT (–) (*n* = 223)	mRDT (+) (*n* = 107)	Total (*n* = 330)	*p*-Value
Age (years)				0.3663 ^1^
Mean (SD)	9.0 (3.1)	8.7 (3.2)	8.9 (3.1)	
Range	1.0–15.0	1.0–15.0	1.0–15.0	
Median	9.0	9.0	9.0	
95% CI	(8.6; 9.4)	(8.1; 9.3)	(8.6; 9.3)	
Age groups (years)				0.6217 ^2^
1–5	25 (11.2%)	14 (13.1%)	39 (11.8%)	
6–15	198 (88.8%)	93 (86.9%)	291 (88.2%)	
Sex				0.9316 ^2^
Female	101 (45.3%)	49 (45.8%)	150 (45.5%)	
Male	122 (54.7%)	58 (54.2%)	180 (54.5%)	
Body weight (kg)				0.2432 ^1^
Mean (SD)	23.3 (8.8)	22.5 (9.4)	23.0 (9.0)	
Range	7.8–59.9	9.9–56.9	7.8–59.9	
Median	21.3	19.9	20.9	
95% CI	(22.1; 24.4)	(20.6; 24.3)	(22.0; 24.0)	
Body temperature (°C)				0.3086 ^1^
Mean (SD)	36.8 (0.3)	36.8 (0.4)	36.8 (0.3)	
Range	36.5–37.5	36.2–37.5	36.2–37.5	
Median	36.7	36.7	36.7	
95% CI	(36.7; 36.8)	(36.7; 36.9)	(36.7; 36.8)	
Temperature categories				0.0551 ^2^
Correct	179 (80.3%)	74 (69.2%)	253 (76.7 %)	
Low-grade fever	44 (19.7%)	33 (30.8%)	77 (23.3 %)	
Fever	0 (0.0%)	0 (0.0%)	0 (0.0 %)	

^1^ U Mann-Whitney, ^2^ Chi-square.

**Table 2 ijerph-18-00814-t002:** Characteristics of Pygmy children (*n* = 170).

Variables	mRDT (–) (*n* = 101)	mRDT (+) (*n* = 69)	Total (*n* = 170)	*p*-Value
Age (years)				0.0139 ^1^
Mean (SD)	7.8 (2.5)	7.0 (1.5)	7.5 (2.2)	
Range	1.0–15.0	3.0–10.0	1.0–15.0	
Median	8.0	7.0	7.0	
95% CI	(7.3; 8.3)	(6.7; 7.4)	(7.2; 7.8)	
Age groups (years)				0.9545 ^2^
1–5	12 (11.9%)	8 (11.6%)	20 (11.8%)	
6–15	89 (88.1%)	61 (88.4%)	150 (88.2%)	
Sex				0.9692 ^2^
Female	48 (47.5%)	33 (47.8%)	81 (47.6%)	
Male	53 (52.5%)	36 (52.2%)	89 (52.4%)	
Body weight (kg)				0.0038 ^1^
Mean (SD)	18.3 (5.5)	16.1 (3.2)	17.4 (4.8)	
Range	9.1–37.8	10.0–26.3	9.1–37.8	
Median	17.3	15.5	16.3	
95% CI	(17.2; 19.4)	(15.3; 16.9)	(16.7; 18.1)	
Body temperature (°C)				0.6546 ^1^
Mean (SD)	36.8 (0.3)	36.8 (0.4)	36.8 (0.4)	
Range	36.1–37.5	36.3–37.5	36.3–37.5	
Median	36.6	36.7	36.7	
95% CI	(36.7; 36.9)	(36.7; 36.9)	(36.8; 36.9)	
Temperature categories				0.6300 ^2^
Correct	73 (72.3%)	48 (69.6%)	121 (71.2%)	
Low-grade fever	28 (27.7%)	21 (30.4%)	49 (28.8%)	
Fever	0 (0.0%)	0 (0.0%)	0 (0.0%)	

^1^ U Mann-Whitney, ^2^ Chi-square.

**Table 3 ijerph-18-00814-t003:** Regression analysis, Bantu children (*n* = 330).

Regression Analysis	Univariate		Multivariate	
	OR (95% CI)	*p*-Value	OR (95% CI)	*p*-Value
Sex				
Females	1.02 (0.64; 1.62)	0.9316		
Males	0.98 (0.62; 1.56)	0.9316		
Age (years)	0.97 (0.90; 1.04)	0.3910		
Age categories (years)				
1–5	1.19 (0.59; 2.40)	0.6220		
6–15	0.84 (0.42; 1.69)	0.6220		
Body weight (kg)	0.99 (0.96; 1.02)	0.4471		
Hemoglobin level (g/dL)	0.86 (0.75; 0.99)	0.0410	0.88 (0.76; 1.02)	0.0805
Anemia categories				
Severe	0.96 (0.35; 2.60)	0.9354		
Moderate	1.47 (0.88; 2.45)	0.1447		
Mild	0.97 (0.61; 1.55)	0.9115		
Normal Hb level	0.61 (0.32; 1.18)	0.1402		
Body temperature (°C)	1.42 (0.73; 2.77)	0.3036		
Body temperature categories				
Correct	0.55 (0.33; 0.93)	0.0266	0.23 (0.02; 2.59)	0.2343
Low-grade fever	1.71 (1.00; 2.91)	0.0496	0.38 (0.03; 4.36)	0.4349

**Table 4 ijerph-18-00814-t004:** Regression analysis, Pygmy children (*n* = 170).

Regression Analysis	Univariate		Multivariate	
	OR (95% CI)	*p*-Value	OR (95% CI)	*p*-Value
Sex				
Females	1.01 (0.55; 1.87)	0.9692		
Males	0.99 (0.54; 1.82)	0.9692		
Age (years)	0.84 (0.72; 0.98)	0.0233	1.05 (0.81; 1.35)	0.7363
Age categories (years)				
1–5	0.97 (0.34; 2.52)	0.9545		
6–15	1.03 (0.40; 2.66)	0.9545		
Body weight (kg)	0.89 (0.82; 0.97)	0.0042	0.88 (0.77; 1.00)	0.0436
Hemoglobin level (g/dL)	0.88 (0.71; 1.08)	0.2264		
Anemia categories				
Severe	2.01 (0.71; 5.70)	0.1867		
Moderate	0.90 (0.47; 1.72)	0.7560		
Mild	1.19 (0.64; 2.20)	0.5837		
Normal Hb level	0.38 (0.12; 1.22)	0.1034		
Body temperature (°C)	1.08 (0.46; 2.54)	0.8643		
Body temperature categories				
Correct	0.88 (0.45; 1.72)	0.7015		
Low-grade fever	1.20 (0.61; 2.36)	0.5987		

**Table 5 ijerph-18-00814-t005:** Prevalence of anemia in the tested Bantu children (*n* = 330).

Variables	mRDT (–) (*n* = 223)	mRDT (+) (*n* = 107)	Total (*n* = 330)	*p*-Value
Hemoglobin level (g/dL)				0.0399 ^1^
Mean (SD)	10.7 (1.7)	10.3 (1.5)	10.6 (1.6)	
Range	5.1–14.7	6.7–13.5	5.1–14.7	
Median	10.9	10.3	10.7	
95% CI	(10.5; 10.9)	(10.0; 10.6)	(10.4; 10.8)	
Anemia categories				0.3315 ^2^
Severe	13 (5.8%)	6 (5.6%)	19 (5.8%)	
Moderate	52 (23.3%)	33 (30.8%)	85 (25.8%)	
Mild	114 (51.1%)	54 (50.5%)	168 (50.9%)	
Normal Hb level	44 (19.7%)	14 (13.1%)	58 (17.6%)	

^1^ t-Student, ^2^ Chi-sqare.

**Table 6 ijerph-18-00814-t006:** Prevalence of anemia in the Pygmy children tested with mRDT (*n* = 170).

Variables	mRDT (–) (*n* = 101)	mRDT (+) (*n* = 69)	Total (*n* = 170)	*p*-Value
Hemoglobin level (g/dL)				0.2269 ^1^
Mean (SD)	10.2 (1.5)	9.9 (1.5)	10.1 (1.5)	
Range	7.1–13.8	5.4–12.6	5.4–13.8	
Median	10.4	10.0	10.2	
95% CI	(9.9; 10.5)	(9.6; 10.3)	(9.9; 10.3)	
Anemia categories				0.2239 ^2^
Severe	7 (6.9%)	9 (13.0%)	16 (9.4%)	
Moderate	36 (35.6%)	23 (33.3%)	59 (34.7%)	
Mild	44 (43.6%)	33 (47.8%)	77 (45.3%)	
Normal Hb level	14 (13.9%)	4 (5.8%)	18 (10.6%)	

^1^ t-Student, ^2^ Chi-square.

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
