# Peer review of "Prevalence of Asymptomatic Malaria Infections in Seemingly Healthy Children, the Rural Dzanga Sangha Region, Central African Republic"

_ijerph, 2021, doi:10.3390/ijerph18020814_

Round 1
Reviewer 1 Report
The study is of interest and an addition to available data in the context of asymptomatic malaria prevalence in Africa and more specifically in CAR.
There is lack of accuracy in the methods and therefore the results given that there is currently ongoing PfHRP2 deletion in circulating parasites. Even if this could be handled by microscopy, the fact that this diagnosis technic depends so much on the microscopist competence is already a hudge bias. What could be suggested is the addition of PCR in order cross-check the found malaria asymptomatic prevalence.
Moreover, given that there are upcoming reports of non-falciparum malaria in neighbouring countries such as Cameroon, there could also be a proportion of non-falciparum malaria among asymptomatic individuals.
Author Response
Correction: Revised as requested
Line 285-303 (additional text):
The fact that WHO recommends simplified diagnostic procedures, i.e. the use of RDTs capable of detecting only one Plasmodium species (P. falciparum) may become a major obstacle to effective malaria control in Africa. According to WHO, P. falciparum is the etiological factor of up to 100% malaria cases in sub-Saharan Africa, but it must be emphasized that a large proportion of cases that occur in the region are never diagnosed because of widespread asymptomatic carriage of Plasmodium.
Unpublished findings of a study which was carried out by the authors in the Bayanga subprefecture (in the same region where the present study was conducted) in 2018, using molecular methods (PCR), in a group of 540 Pygmies (aged 1-75 years) presenting with clinical signs of malaria, confirmed the presence of P. falciparum in 94.8% patients, but also found genetic material of P. malariae in 11.1%, P. ovale in 9.8%, and P. vivax in 0.7% study subjects. The results obtained indicate that there are different species of Plasmodium in Central Africa, which is a significant finding with respect to implementing effective diagnostic procedures and treatment protocols (currently, it is quite common that patients with a negative result on mRDT which is specific for detecting P. falciparum only, do not receive antimalarial drugs although they manifest clear signs of the infection).
One should also be aware of numerous limitations of RDTs (poor sensitivity at low parasite densities, susceptibility to the prozone effect /Pf HRP2-detecting RDTs/), the possibility of false negative results (due to Pf HRP2 deficiency in the case of pfhrp2 gene deletions /Pf HRP2-detecting RDTs/ and cross-reactions between Plasmodium antigens and detection antibodies), as well as the possibility of false-positive results caused by other infections [22].
Reviewer 2 Report
The manuscript by Korzeniewski et al investigates the prevalence of asymptomatic malaria infections in a region of the central african republic (CAR). The data show convincingly high prevalence of Plasmodium falciparum infection rates in children of settled Bantu as well as in semi-nomadic living BaAka pygmies. The authors discuss possible reasons for the high prevalence of infection as well as consequences regarding public health.
The manuscript is written in decent and concise english and data are presented in a clean and precise way. The overall topic of the study is interesting since it is still under debate why some infections remain asymptomic while other patients develop clinical symptoms. More and better field data may help to find pattern in the occurrence of asymptomatic infections which might help to understand underlying host-pathogen mechanisms.
Minor edits:
- (Line 53-56) As far as I know is the lower mortality related to COVID-19 in african countries mainly based on the lower age average. Younger people often develop an asymptomatic or a less severe course of the disease leading to less deaths. However, this doesn’t mean that SARS-CoV-2 isn’t highly prevalent in the population which could be a potential risk for tourists. Moreover are the auhors questioning the published data for Plasmodium infections by the government of CAR and estimate that malaria prevalence is possibly much higher. I would assume that the same could be true for COVID-19 cases making it even more difficult to argue with given numbers. This passage should be completely removed from the manuscript.
- (Line 360-362) This sentence should be removed for the same reasons as explained above.
- (Line 41-43) Sporozoites are mostly deposited in the skin by the biting mosquito and need to migrate through the dermis to find and invade blood capillaries (Douglas et al., 2015). Please rephrase this sentence.
- (Line 43-45) Transmission by blood transfusion and vertical transmission of Plasmodium can occur. However, this a rather rare event and authors do not comment on this later in the manuscript. Moreover will be any asexual blood stage present in transfused blood cause an infection and not just trophozoites or schizonts. I would recommend to rephrase or completely remove this sentence.
- (Line 131) It should read RDT instead of RTD.
- The authors discuss their findings in relation to other studies in the area and neighbouring countries. Still, it would have helped if the authors would have deepened the discussion about the reasons for the observed high asymptomatic prevelance of malaria infection. For example it would have been interesting to discuss the prevalence of asymptomatic infections in relation to symptomatic infections as well as malaria caused fatalities in the area. This could point to an imbalance between different disease courses which might give hinds about underlying mechanisms.
General comments and recommendations
- The ° in degree celsius is always placed in the middle but should be placed higher (°C). Probably a formatting issue.
- (Line 26-28) Average age, hemoglobin level and body temperature of the participants are given in the abstract. In my opinion this is wrongly placed and could be deleted to make the abstract more concise.
- (Line 22) The description of the used RDT is kind of long and cumbersome phrased. Could possibly be simplified.
- Its sometimes very difficult to compare values within different tables in the result section which requires a lot of scrolling. In my opinion the presentation of data could have been improved if some tables were presented in a graphical way and only really relevant data were incorporated in the main manuscript while less discussed data could have been moved to the supplement.
References
Douglas, R. G. et al. (2015) ‘Active migration and passive transport of malaria parasites’, Trends in Parasitology, pp. 357–362. doi: 10.1016/j.pt.2015.04.010.
Author Response
Answers to the Reviewer 2
Line 53-56: As far as I know is the lower mortality related to COVID-19 in african countries mainly based on the lower age average. Younger people often develop an asymptomatic or a less severe course of the disease leading to less deaths. However, this doesn’t mean that SARS-CoV-2 isn’t highly prevalent in the population which could be a potential risk for tourists. Moreover are the auhors questioning the published data for Plasmodium infections by the government of CAR and estimate that malaria prevalence is possibly much higher. I would assume that the same could be true for COVID-19 cases making it even more difficult to argue with given numbers. This passage should be completely removed from the manuscript.
Correction: Done
Line 360-362: This sentence should be removed for the same reasons as explained above.
Correction: Done
Line 41-43: Sporozoites are mostly deposited in the skin by the biting mosquito and need to migrate through the dermis to find and invade blood capillaries (Douglas et al., 2015). Please rephrase this sentence.
Douglas, R.G; Amino, R.; Sinnis, P.; Frischknecht, F. Active migration and passive transport of malaria parasites. Trends Parasitol. 2015, 31, 357–62. [https://doi.org/10.1016/j.pt.2015.04.010]
Correction: Done
Line 43-45: Transmission by blood transfusion and vertical transmission of Plasmodium can occur. However, this a rather rare event and authors do not comment on this later in the manuscript. Moreover will be any asexual blood stage present in transfused blood cause an infection and not just trophozoites or schizonts. I would recommend to rephrase or completely remove this sentence.
Correction: Revised as requested
Line 131: It should read RDT instead of RTD.
Correction: Done
The authors discuss their findings in relation to other studies in the area and neighbouring countries. Still, it would have helped if the authors would have deepened the discussion about the reasons for the observed high asymptomatic prevelance of malaria infection. For example it would have been interesting to discuss the prevalence of asymptomatic infections in relation to symptomatic infections as well as malaria caused fatalities in the area. This could point to an imbalance between different disease courses which might give hinds about underlying mechanisms.
Correction: Revised as requested
Line 283 (additional text):
The fact that WHO recommends simplified diagnostic procedures, i.e. the use of RDTs capable of detecting only one Plasmodium species (P. falciparum) may become a major obstacle to effective malaria control in Africa. According to WHO, P. falciparum is the etiological factor of up to 100% malaria cases in sub-Saharan Africa, but it must be emphasized that a large proportion of cases that occur in the region are never diagnosed because of widespread asymptomatic carriage of Plasmodium.
Unpublished findings of a study which was carried out by the authors in the Bayanga subprefecture (in the same region where the present study was conducted) in 2018, using molecular methods (PCR), in a group of 540 Pygmies (aged 1-75 years) presenting with clinical signs of malaria, confirmed the presence of P. falciparum in 94.8% patients, but also found genetic material of P. malariae in 11.1%, P. ovale in 9.8%, and P. vivax in 0.7% study subjects. The results obtained indicate that there are different species of Plasmodium in Central Africa, which is a significant finding with respect to implementing effective diagnostic procedures and treatment protocols (currently, it is quite common that patients with a negative result on mRDT which is specific for detecting P. falciparum only, do not receive antimalarial drugs although they manifest clear signs of the infection).
One should also be aware of numerous limitations of RDTs (poor sensitivity at low parasite densities, susceptibility to the prozone effect /Pf HRP2-detecting RDTs/), the possibility of false negative results (due to Pf HRP2 deficiency in the case of pfhrp2 gene deletions /Pf HRP2-detecting RDTs/ and cross-reactions between Plasmodium antigens and detection antibodies), as well as the possibility of false-positive results caused by other infections [22].
The ° in degree celsius is always placed in the middle but should be placed higher (°C). Probably a formatting issue.
Remark: unfortunately, it is related to the formatting of the file
Line 26-28: Average age, hemoglobin level and body temperature of the participants are given in the abstract. In my opinion this is wrongly placed and could be deleted to make the abstract more concise.
Correction: Done
Line 22: The description of the used RDT is kind of long and cumbersome phrased. Could possibly be simplified.
Correction: Done
Its sometimes very difficult to compare values within different tables in the result section which requires a lot of scrolling. In my opinion the presentation of data could have been improved if some tables were presented in a graphical way and only really relevant data were incorporated in the main manuscript while less discussed data could have been moved to the supplement.
Correction: Revised as requested; tables 2,3,6,7 have been removed
Reviewer 3 Report
In this paper the prevalence of asymptomatic malaria infections of seemingly healthy children living in the rural parts of the Dzang Sangha region in south-west Central African Republic, and the results is valuable for malaria prevention and control strategies in Central African Republic. Overall, the results are clearly stated and the methods selected were appropriate for the questions presented. If authors could revise the manuscript well, I will support the publication of the work in Frontiers. Specific comments to follow: 1 the title of manuscript is” Prevalence of Asymptomatic Malaria Infections in 2 the Rural Dzanga Sangha Region”, however the author only focus on assessing the prevalence of asymptomatic malaria infections of seemingly healthy children. So ,we suggest rewrite the title. . 2 We suggest author to report the actual infection rate of malaria in the Bantu and Pygmy children aged 1–15 years, and compare the Asymptomatic Malaria Infections rate and actual infection rate, to further discuss the reason in discussion section 3 according to The exclusion criteria “presence of 97 malaria signs and symptoms (fever ≥37.6 o C, chills, headache, vomiting, diarrhea, joint pain, general weakness), anti-malarial treatment received in the past 28 days and difficulties in venipuncture procedure. Considering the influence of other epidemic disease which cause the similar symptoms, we suggest the author add the description about Local epidemic profiles in discussion section.Author Response
Answers to the Reviewer 3
The title of manuscript is ”Prevalence of Asymptomatic Malaria Infections in the Rural Dzanga Sangha Region”, however the author only focus on assessing the prevalence of asymptomatic malaria infections of seemingly healthy children. So, we suggest rewrite
the title.
Lines 2-3 Correction of the manuscript title:
Prevalence o Asymptomatic Malaria Infections in Seemingly Healthy Children, the Rural Dzanga Sangha Region, Central African Republic
We suggest author to report the actual infection rate of malaria in the Bantu and Pygmy children aged 1–15 years, and compare the Asymptomatic Malaria Infections rate and actual infection rate, to further discuss the reason in discussion section 3 according to the exclusion criteria (Line 97) “presence of malaria signs and symptoms (fever ≥37.6oC, chills, headache, vomiting, diarrhea, joint pain, general weakness), anti-malarial treatment received in the past 28 days and difficulties in venipuncture procedure. Considering the influence of other epidemic disease which cause the similar symptoms, we suggest the author add the description about local epidemic profiles in discussion section.
Correction: Revised as requested
Discussion Line 318-328 (additional text):
The results of this study, which was aimed to assess the occurrence of asymptomatic malaria infections in a sample of 500 children, aged 1-15 years, living in the rural areas of the Dzanga Sangha region in the south-west of the CAR, revealed 35.2% of asymptomatic malaria cases: 32.4 % in Bantu and 40.6 % in Pygmy children.
A total of 516 children entered the study (339 Bantu and 177 Pygmy), of which 16 were excluded because of fever ≥37.6oC (their parents/ legal guardians did not report of any other signs and symptoms of malaria, such as chills, headache, vomiting, diarrhea, joint pain, general weakness; nor did they report of anti-malarial treatment received by their children in the past 28 days; no difficulties occurred in venipuncture procedure). Of the 16 children with fever ≥37.6oC (9 Bantu and 7 Pygmies) who were excluded from the study, a majority presented with symptoms of a respiratory infection (cough, rhinitis); the possibility of other febrile illnesses, that are epidemic (dengue) or endemic (tuberculosis, lymphatic filariasis caused by Wuchereria bancrofti ) in Central Africa, that have not been studied in the local population, should also be considered. The present study is the first screening ever conducted in asymptomatic patients with malaria infection in the Dzanga Sangha region, in south-west CAR.
Round 2
Reviewer 1 Report
There is improvement in the new version of the manuscript.
Nevertheless, I doubt the ethical consent of the individuals participating in this study is sufficient to cover the IRB requirements in CAR. It is not appropriate to have a study carried out in a country with an IRB approval from a foreign country. There was a need to seek a local IRB approval before initiating the survey.
Author Response
Line 151-152:
The research project was approved by the Ministry of Scientific Research and Technological Innovation, Bangui, Central Africa Republic (Research Authorization)
